# Divergent Trends in the Prevalence of Children’s Asthma, Rhinitis and Atopic Dermatitis and Environmental Influences in the Urban Setting of Zagreb, Croatia

**DOI:** 10.3390/children9121788

**Published:** 2022-11-22

**Authors:** Iva Topalušić, Asja Stipić Marković, Marinko Artuković, Slavica Dodig, Lovro Bucić, Liborija Lugović Mihić

**Affiliations:** 1Division of Pulmology, Immunology, Allergology and Rheumatology, Department of Paediatrics, University Children’s Hospital Zagreb, 10 000 Zagreb, Croatia; 2Department of Pulmology, Special Hospital for Pulmonary Diseases, 10 000 Zagreb, Croatia; 3Faculty of Pharmacy and Biochemistry, University of Zagreb, 10 000 Zagreb, Croatia; 4Division for Environmental Health, Croatian Institute for Public Health, 10 000 Zagreb, Croatia; 5Department of Dematology, School of Dental Medicine, Clinical Hospital Center Sisters of Mercy, 10 000 Zagreb, Croatia

**Keywords:** children’s allergy rise, allergic rhinitis, atopic dermatitis, urban asthma, exposome

## Abstract

Background: Previous studies have reported that the allergy epidemic in developed countries has reached its plateau, while a rise is expected in developing ones. Our aim was to compare the prevalence of allergic diseases among schoolchildren from the city of Zagreb, Croatia after sixteen years. Methods: Symptoms of asthma, allergic rhinitis (AR) and atopic dermatitis (AD) and risk factors were assessed using the International Study of Asthma and Allergies in Childhood (ISAAC) questionnaire. An allergic profile was determined by a skin prick test. Results: The prevalence of current, ever-in-a-lifetime and diagnosed AR of 35.7%, 42.5% and 14.9% and AD of 18.1%, 37.1% and 31.1% demonstrated a significant increase. The asthma prevalence has remained unchanged. The allergen sensitivity rate has remained similar, but pollens have become dominant. Mould and dog exposure are risks for asthma (OR 14.505, OR 2.033). Exposure to cat allergens is protective in AR (OR 0.277). Parental history of allergies is a risk factor in all conditions. Conclusion: Over sixteen years, the prevalence of AR and AD, but not of asthma, have increased. The proportion of atopy has remained high. The AR/AD symptom rise is probably a consequence of increased pollen sensitisation united with high particulate matter concentrations. The stable asthma trend could be a result of decreasing exposures to indoor allergens.

## 1. Introduction 

Allergies are a topic of international concern with a great disparity over time and geographical regions as a consequence of complex interactions between genes specific to a particular population as well as rapidly changing environmental exposures in developed societies [1,2,3]. However, important distinctions between specific elements responsible for the allergy pandemic are still difficult to establish [4,5]. The allergy pandemic was a driving force for the establishment of standardised research methodologies promoted by the International Study of Asthma and Allergies in Childhood (ISAAC) in 1992, with the aim of a comparing of results and follow-ups of temporal trends across geographical regions [6,7,8]. Current evidence regarding time trends in allergies appears to be limited [9,10,11,12]. In the school year of 2001/2002, a few years after the first ISAAC study, which demonstrated the highest prevalence in the most developed countries among 56, ISAAC Phase One was conducted in the city of Zagreb, the capital of the low-income Central European country of Croatia [13]. This study shows the prevalence of asthma, allergic rhinitis (AR) and atopic dermatitis (AD) in ranges II and III out of IV (I < 5%; II 5 to <10%; III 10 to <20%; IV ≥ 20%) [13]. A similar prevalence was also found in other regions of the country [14]. The identification of risk factors in allergic participants showed a statistically higher rate of day care attendance and pertussis infections in comparison to nonallergic children, while exposure to tobacco smoke during pregnancy as well as indoor dampness and mould also led to risks for allergic diseases [15]. Sensitisation to common aeroallergens was detected in 52.3% of children, with house dust mites being the most dominant [16]. The aim of this study, conducted in the year 2017/2018, is to compare the prevalence of allergic symptoms and the role of environmental exposure in the disease onset, with the results of this study conducted in the school year of 2001/2002 using the same methodology. These estimates could be helpful for future preventive strategies regarding the window of opportunity for life-long immune homeostasis [17].

## 2. Materials and Methods 

A total of 1047 children in the school year of 2001/2002 and 454 children in the school year of 2017/2018, aged 10–11, took part in this study. This study was conducted among 18 randomly selected schools in the city of Zagreb in the school year of 2001/2002 and 30 schools in the repeated study. The sample size was calculated according to ISAAC recommendations [6] of at least 1000 participants in the study group in the school year of 2001/2002. The exceptions are smaller populations, where the sample size of 1000 participants could not be reached, which was the case in the repeated study, due to the very low participation rate (50%). The response rate in the first ISAAC study was higher, about 80%.

Standardised and validated ISAAC questionnaires were used and answered by parents. As epidemiological definitions of asthma, we used questions on symptoms of wheezing ever-in-a-lifetime and during the last 12 months (current wheezing), and a third, more specific, but less sensitive question was on self-reported diagnosed asthma. As the definition of AR, questions on a runny nose without a cold ever-in-a-lifetime and during the last 12 months (current AR) as well as a question on self-reported diagnosed AR were used. Itchy rashes, both ever-in-a-lifetime and during the last 12 months (current AD), as well as a question on self-reported diagnosed AD were used as definitions of this disorder (Table 1). We also analysed allergic multimorbidities.

We also used a set of questions on environmental exposure and family history. The questions on environmental exposure consisted of:Questions on early-life exposure (attending day care, breastfeeding, pertussis vaccination, smoking during pregnancy and parasite infestations);Exposure that influences the microbiome (dog and cat exposure in the first year of life and during the last 12 months);Exposure to the indoor environment (mould and dampness in the first year of life and during the last 12 months, using feather pillows during the last 12 months and the frequency of hamburger consumption during the last 12 months).

The studied environmental factors are shown in Table 2 and Table 3.

The questions on family history consisted of questions on the father’s asthma, atopic dermatitis and allergic rhinitis as well as on the mother’s asthma, atopic dermatitis and allergic rhinitis. 

Among the environmental exposures assessed by the questionnaire, we compared the prevalence of those that were indicated as risks or as protective in the previous investigation in the city of Zagreb [15], and the association of those factors with allergic symptoms was studied in the logistic regression. All the questionnaires were checked, and inconsistent answers were discussed with parents. Skin prick tests (SPT) to common whole aeroallergen extracts (Allergopharma^®^ GmbH & Co. KG, Reinbek, Germany) were conducted on 354 children aged 10 in 2002 and 208 children aged 7–14 in 2018 according to the standardised methodology [18]. The set of the tested allergens included grasses, ragweed, Dermatophagoides pteronyssinus, Dermatophagoides farinae, late blooming trees, early blooming trees and cat dander.

Statistical analyses were completed using the SPSS software (26.0, SPSS Inc., Chicago, IL, USA). Categorical variables were described by frequencies and proportions. Differences in the prevalence of allergic symptoms and environmental exposures over the studied period were tested using the Z-test, with *p* < 0.05 considered statistically significant. Multivariate logistic regression was used to assess associations between environmental factors and allergic symptoms, with the strength of association quantified by odds ratios (OR) and 95% confidence intervals (CI). 

## 3. Results

### 3.1. Demographic Data

A total of 1047 children aged 10–11 from 18 schools were enrolled in the school year of 2001/2002. Their demographic features were published previously [13,16]. In the school year of 2017/2018, 454 children aged 10–11 from 30 elementary schools, 225 girls and 229 boys, were recruited. The summary of the data on their family history and environmental exposures are listed in Table 2.

### 3.2. Current Prevalence of Allergic Symptoms and Comparison with the Year 2002

The self-reported prevalence of current asthma symptoms in 2018 was 5.7%. None of the asthma variables changed significantly in comparison with the results from 2002 (Figure 1).

The estimated self-reported prevalence of current AR in 2018 was 35.7%. The prevalence of AR symptoms ever-in-a-lifetime increased significantly by 28% (*p* < 0.001), current rhinitis by 23% (*p* < 0.001) and self-reported diagnosed AR by 5% in comparison with the previous ISAAC study (*p* = 0.006) (Figure 2).

The self-reported prevalence of current AD in 2018 was 18.1%. The symptoms of AD ever-in-a-lifetime increased by 19% (*p* < 0.001), current AD by 10% (*p* < 0.001) and self-reported AD diagnoses by 20% (*p* = 0.001) in comparison with the previous study (Figure 3).

In regard to the frequency of allergic multimorbidities in 2018 (at least two allergic diseases simultaneously), it was demonstrated that current AR was dominantly a single disease (67% children). Asthma appeared to be the syndrome with the highest rate of multimorbidity (81%), while multimorbidities were also frequent in patients with current AD symptoms (58%) (Figure 4).

### 3.3. Skin Sensitivity to Common Aeroallergens

Among the tested children, 47% of them had at least one positive skin test. The most common allergens were grasses followed by ragweed, Dermatophagoides pteronyssinus, Dermatophagoides farinae, late blooming trees, early blooming trees and cat dander (Figure 5). 

In comparison with the year 2002, the allergen sensitivity rate was similar, while the atopy profile was slightly different, with grasses being a dominant sensitiser. 

### 3.4. Environmental Exposure

Among the environmental exposures assessed by the questionnaire, we compared the prevalence of those that were indicated as risks or as protective in the previous investigation [15]. There has been a significant increase in the number of children attending day care facilities (*p* < 0.001) and getting pertussis vaccinations (*p* = 0.036). Maternal tobacco smoke exposure during pregnancy has also increased significantly (*p* < 0.001). Indoor exposure to dampness during the first year of life and during the last 12 months as well as the exposure to moulds during the last 12 months were more frequent in comparison with the year 2002, as well (*p* < 0.001, *p* < 0.003, respectively). The use of feather pillows has decreased significantly (*p* < 0.001) (Figure 6). 

In order to characterize associations of the disease symptoms and various risk factors, a logistic regression was performed. As independent variables, we studied the risk factors shown in Table 3. The logistic regression for asthma symptoms showed that the mother’s AR (OR 2.813, *p* < 0.001, 95 CI 1.608–4.921) and the father’s AR (OR 2.527, *p* = 0.004, 95% CI 1.338–4.773) as well as currently living with a dog (OR 2.033, *p* = 0.018, 95% CI 1.127–3.667) were positively associated with developing wheezing ever-in-a-lifetime. The mother’s AR (OR 4.167, *p* = 0.018, 95% CI 1.284–13.527) and the presence of mould in home during the first year of life (OR 14.505, *p* = 0.044, 95% CI 1.076–195.524) were risk factors for developing current asthma. The diagnosis of asthma was positively associated with the mother’s AR (OR 3.082, *p* = 0.022, 95% CI 1.178–8.065) (Table 3). In the regression model for AR symptoms, the mother’s AR was also positively associated with AR symptoms ever-in-a-lifetime (OR 1.689, *p* = 0.043, 95% CI 1.018–2.803), while currently living with a cat was negatively associated with current AR (OR 0.277, *p* = 0.030, 95% CI 0.087 −0.883). The diagnosis of AR was positively connected with the mother’s AR (OR 2.371, *p*= 0.008, 95% CI 1.257–4.473) and the father’s AR (OR 2.841, *p* = 0.003, 95% CI 1.414–4.473) (Table 3). 

In the regression model for the symptoms of AD, the mother’s AD and the father’s AR were positively associated with AD symptoms ever-in-a-lifetime (OR 5.124, *p*= 0.000, 95% CI 2.199–11.943; OR 1.983, *p* = 0.026, 95% CI 1.087–3.617) and the diagnosis of AD (OR 2.752, *p* = 0.010, 95% CI 1.275–5.943; OR 2.095, *p* = 0.014, 95% CI 1.159–3.787), while none of the factors were associated with current AD (Table 3).

## 4. Discussion

This comparative study estimated the self-reported prevalence of current asthma in 2018 to be 5.7%. In comparison with the results from the first ISAAC study in 2002, none of the epidemiological definitions of asthma demonstrated a significant increase or reduction over the past decade and a half. It has been acknowledged that the prevalence of childhood asthma may have reached its plateau in developed countries such as Australia and the UK. There are concerns about the rising trends in asthma in middle- and low-income countries, especially in cities [10]. The results of our study do not justify that. However, other European countries are far behind in analysing time trends in allergic diseases. In 2016, activities that were aimed towards devising new data from Poland, Ukraine and Belarus provided a low asthma prevalence of 4.1%, 2.1% and 1.5% in urban regions, respectively [11]. The prevalence of asthma symptoms in Katowice, Poland in 2019 was similar to our results. Wheezing ever was present in 20.8%, wheezing during the last 12 months in 7.2% and self-reported asthma diagnoses in 7.3% of children [19]. The reported prevalence of asthma in Budapest, Hungary in 2020 was 6.5%, which was also consistent with our findings [12]. These results, together with our time trends, may indicate that the asthma gradient assessed in 1998, demonstrating the highest prevalence in the north of Europe (Finland 19%) and declining towards the south (Albania 2.65%), might be constant [7]. 

Atopy and an increased number of IgE recognition frequencies are considered as strong risks for the asthma phenotype in children [20]. Atopy and asthma share some single-nucleotide polymorphisms (SNPs), mainly related to immune interactions with endotoxins [21]. Progression from atopy as a predisposition towards the clinical symptoms of asthma starts with exposure to indoor allergens, followed by a sensitisation phase of the allergic reaction. These relationships have been well documented in young and school-aged children, in particular for house dust mites [22,23]. Atopy and sensitisation in early life had greater impacts on asthma development than sensitisation later in childhood. This was shown in the Multicentre Asthma Study where only 56% of atopic children had remission of wheezing symptoms by the age of 13 (approximately the age of our study population) in comparison with 100% remission in children with wheezing symptoms without sensitisation [24]. In our study, as in some other studies [12,25], the prevalence of atopy of 47% was much higher than the asthma prevalence, illustrating that allergen sensitisation did not lead to the development of asthma symptoms later in life. 

We speculate that allergen remediation strategies directed to indoor allergens substantially decreased their content in homes and reduced or delayed the onset of asthma. In the city of Zagreb, decades of preventive work through 61 out-clinic paediatric offices and two paediatric hospital departments has been conducted with organised asthma schools for parents and children, collaborating with pulmonary rehabilitation centres on the seaside [26,27,28,29,30,31,32,33]. Consistent with these observations are meta-analyses which have shown multifaceted intervention programs to be protective against the development of asthma, especially in a high-risk birth cohort [34,35,36] but did not show a significant decrease in atopy at the 8-year time point in other studies [37,38]. Compared with the year 2002, the sensitisation profile in our study population was slightly shifted to grass pollens and ragweed, which provide repeated outdoor allergen stimulation and are less important in asthma development than perennial allergens. 

One of the indoor allergens significantly associated with current asthma symptoms in our study was the dog allergen. Although some studies have shown a protective effect of early-life exposure to dogs on asthma development, there have been some opposite results [39,40,41]. A recent meta-analysis has found a protective effect of dog exposure on asthma in birth, but not in nonbirth cohorts, e.g., in adolescents [39]. It has been reported that the protective effect depends on the number of dogs and their sex [42] as well as the period of exposure and amount of allergen [39]. 

Another indoor allergen significantly associated with current asthma symptoms in our study was mould, as has been shown in the BUPAS study [11] and in huge meta-analyses on the associations of respiratory health with dampness and mould in homes [43,44]. A higher exposure to mould could be a result of socioeconomic differences between the studied populations. However, climate change may increase the prevalence of dampness and indoor and atmospheric mold exposure [45]. 

Other asthma-related single exposures, such as maternal smoking during pregnancy and indoor dampness, that have been shown to interact with asthma-candidate SNPs leading to antioxidative impairment [46,47,48,49,50,51] and which have increased significantly during the last 16 years [52,53], were not asthma risks according to our regression analysis. 

As well as in other studies, our research shows that a family history of allergies is a risk factor contributing to higher odds of all three asthma definitions [54,55,56]. However, as commented in the paper by Krautenbacher and colleagues, family history may reflect not only classical inheritance, but also a shared environment, including epigenetic mechanisms [57]. The stable asthma trends in our investigation should be observed in the context of the balance between the protective (e.g., microbial load, endotoxins) factors (mostly from atopy) and risk factors (e.g., air pollution, smoking) together with their interactions with the susceptible genes in our population. Genome-wide association studies (GWAS) could help in better understanding these interactions and the observed stable time trends of asthma symptoms.

In contrast to asthma, our study showed an exponential increase in all variables of AR during the last decade and a half. The estimated prevalence of the current AR of 35.7% has reached the prevalence of the most developed countries [7,8], which is consistent with some other Central European countries [12,19]. The study from Budapest, Hungary showed a current AR prevalence of 29.3%, a physician-diagnosed AR prevalence of 9.7% and a cumulative AR prevalence of 36.2% [12]. In Katowice, Poland, the prevalence of self-reported AR (25.8%) and diagnosed AR (22%) were also high [19]. Consistent with these findings are the results of Ha and co-workers on the increase in the AR prevalence rate of up to 27.6% in the population of 12 919 Korean school-aged children (6 to 18 years) during a period of 10 years [58], as well as results from 11 major cities in China [59,60]. 

Similar to the asthma results, the parental history of AR in our patients suggested a genetic component, which is known to occur in allergic diseases but could not be the cause of such rapid changes in the allergy epidemic [61,62,63]. Therefore, we expected the increase in AR symptoms to be explained by environmental and social changes. We showed a significant increase in several single-environmental exposures in the observed time period: day care attendance, maternal tobacco smoking during pregnancy, indoor dampness and moulds, pertussis vaccinations and cat exposure. However, the association was statistically significant only for cat allergens as a protective factor, with the strength of the association of OR of 0.277 and a 95% CI of 0.087–0.883. Cat allergens have unique properties. The major allergen, Fel d 1, is the most spread indoor animal allergen in Central Europe [64]. It is present not only in homes, but also in public places such as schools. The balance between sensitisation and tolerance is not fully understood, although it has been shown that tolerogenic IgG4 antibodies have been found in cat owners [65]. The “tolerance” to cat allergen exposure achieved in early childhood can be maintained for a long period of time [66]. 

As a possible factor contributing to the increasing prevalence of AR in our city, we speculate that the increasing air pollution, such as in Budapest, Hungary and the Chinese Dejang region, plays a role [12,60,67]. According to data from the Croatian Institute for Public Health and the City of Zagreb, air concentrations of particulate matter, (PM)10 and NO_2_, were above the accepted in the year 2018. Air pollution was a reason for the establishment of the action plan for improving air quality in 2015 [52]. Nasal epithelial cells from subjects with AR released significantly greater amounts of IL-8 and RANTES in response to diesel particles, as shown in the study of Ozturk and co-workers [68]. Furthermore, even low concentrations of NO_2_ have a synergistic effect with allergenic compounds on proinflammatory nasal pathways [69]. It has been acknowledged that air pollution has a role in the fragmentation of pollens and their penetration to the airways, thus contributing to the allergic sensitisation and aggravation of symptoms [70]. 

In our study, the slight shift in the skin test positivity from dominant mites in 2002 towards grasses and ragweed in 2018 could be consistent with this observation, contributing to the rising trend of AR. In addition, according to our previously published data, AR symptoms in children in the city of Zagreb are most prominent in August and September, during the period of ragweed pollination [71], as has been demonstrated in Budapest, Hungary [12]. Taking into account a peak in AR symptoms during the pollen season, the shift in skin positivity from mites to pollen allergens while the total percent of sensitised children remained unchanged, and finally, the increasing concentration of PM10 in children’s urban settings, it can be assumed that the rising trend of AR is a result of a specific combination of these factors, which can be defined as the exposome [72]. The risk effect of air pollution on AR has recently been demonstrated in multi-exposure models [73]. The increase in the AR prevalence could be the result of the appearance of clinical symptoms in previously asymptomatic patients after the abovementioned specific stimulation. A longer pollen season and a higher allergen load could stimulate the activation of nasal epithelial cells, pushing the allergic reaction from the sensitisation phase to effector stage [67,68]. Why this exposure has not influenced distal airways needs to be clarified. In the study of Krmpotić and co-workers, levels of PM10 did not influence the rate of asthma hospitalisations in adults in the city of Zagreb either [74]. 

In our population, 67% of children with AR had only nasal symptoms (Figure 4), which was consistent with the known observation that allergic rhinitis more commonly appears as a single disease in comparison to asthma and AD [75]. This compartmentalisation of allergic inflammation could be underlined with the findings of new GWASs, in which it has been discovered that allergic multimorbidities and single allergic diseases do not have a common genetic base [75]. 

In our urban cohort, the AD frequency increased significantly during the sixteen-year period, reaching the prevalence of the most developed countries [7,8]. The estimated prevalence of AD ever-in-a-lifetime of 36.6%, of current AD of 18.1% and of diagnosed AD of 31% were similar to the results from surrounding Central European countries, such as the urban setting of Poland (20.5% for diagnosed AD) and Budapest, Hungary (12.8%) [12,19]. In the Eastern European country of Romania, the AD prevalence increased from 11.5% to 16.2% in 13-year-old children from 1995 to 2001 [76]. 

A strong risk factor for the disease in our study, as expected, is a history of allergies in both parents. The history of the mother’s AD is associated with the child’s AD ever-in-a-lifetime with a strength of association OR of 5.124 and a 95% CI of 2.199–11.943, and the father’s AR history is associated with an OR of 1.983 and a 95% CI of 1.087–3.617. Many studies have underlined the role of family history in the development of AD. Roduit and co-workers showed that children’s risk of developing the early-persistent AD phenotype is five times greater in the case of parental allergies [77]. In addition, a birth cohort in Switzerland demonstrated that family history is the major risk factor for AD in Caucasians, regardless of even the most common filaggrin mutation, which was infrequent in this cohort [78]. 

In regard to the environmental factors recognised which disrupt the epithelial barrier, we analysed smoking, mould and dampness in homes, which increased in frequency in comparison to the study from 2002 (Figure 6), but we did not find any association with the onset of AD by logistic regression analysis. This result, different from the studies of other ethnic cohorts [79,80,81]_,_ may be due to the absence of the filaggrin gene mutation in Zagreb adolescents that could contribute to epithelial barrier protection from environmental influences [82,83]. Another ambient pollution known to influence AD is PM10 [84]. PM10 exacerbates AD in sensitised mice by recruiting mast cells in the dermis, elevating the serum IgE level and stimulating the expression of proallergic genes (e.g., IL-33) not only in inflamed, but also in intact skin [85]. We could not exclude the effect of increased PM10 concentrations in Zagreb [51,52] on the progression of the disease from predisposition to clinical relevance. 

Another provocative factor for AD is food. Western nutritional habits such as fast food consumption influence the onset of AD, as shown in the South African Food Allergy (SAFFA) cohort study [86]. The ISAAC ecological analysis also demonstrated that the consumption of hamburgers more than three times per week increases the risk of AD [87]. However, in our study population, the frequency of hamburger consumption was low with only 14% of children consuming hamburgers less than once a week and only 0.6% more than three times a week, which seemed not to be relevant for the increasing AD trend. 

Usually, AD is considered the starting point of the “atopic march”, followed by allergic rhinitis and asthma [88]. In this study, it led to the increase in AR only, while asthma symptoms were unaffected. This is consistent with the observation that the European ancestry of genetic polymorphisms is more often associated with the progression from eczema to rhinitis, rather than from eczema to asthma [89]. GWASs have shown that single allergic diseases and allergic multimorbidities have a different genetic base, which could mean that the classical “atopic march” represents only part of the pool of allergic patients with a common genetic basis [74]. Progression from one atopic disease is also geographically determined, as previously mentioned. On the other hand, the “atopic march” progression depends on the phenotypes of AD [89], which cannot be captured by an ISAAC questionnaire alone. Further gene–environmental studies and latent class analysis studies could offer a better explanation for these divergent findings.

Limitations of this study: The discrepancy in the number of participants between the first and second study is a possible methodological weakness of this study. We suppose that the reasons for such a low response rate were social changes during the 16-year period regarding the modern, busy lifestyle, leading to the parents becoming less willing to participate in this study. The fact that the parents of children suffering from allergic diseases are more motivated to participate in this study than the parents of healthy children is a problem for this kind of methodology in general. However, our larger, still not published study in the city of Zagreb, conducted in three age groups: 7, 10 and 14 years old, with a total number of 1099 children, shows a similar total prevalence of asthma symptoms as in this subgroup of 454 10-year-old children. The methodological discrepancy between the first and the second study related to the age of subjects who were skin prick tested could also affect the final conclusions. It is known that the prevalence of skin sensitization to inhalant allergens increases with the age of children. Taking into account the fact that in this study, we included also teenagers, we can conclude that the real prevalence of skin sensitization is certainly not higher than in the previous investigation. 

## 5. Conclusions

The frequency of AR and AD in a 10-year-old urban population of the low-income country of Croatia increased to a high proportion of 35.7% for current AR and 18.1% for current AD symptoms. The multivariate logistic regression analysis failed to confirm associations of these diseases with lifestyle variables, such as tobacco smoking during pregnancy, day care attendance and indoor mould and dampness but, at the same time, showed an inverse correlation between AR and exposure to cat allergens (OR 0.277, 95% CI 0.087–0.883). The proportion of sensitised children of 47% has not changed over time, although a shift in the sensitisation profile from mites to pollens has been registered. We speculate that the increased level of sensitisation to pollen along with the report of increased concentrations of particulate matter in the urban atmosphere could be the cause of the increased AR prevalence. We have not shown any associations between environmental exposure and AD in the population in which filaggrin mutations were not found. A child’s genetic background is confirmed in both disorders, but in AD, it is related to both parents. Asthma is related to a parental history of allergies, exposure to dogs and indoor mould. The mechanisms responsible for the stable asthma trends in our population could be explained by preventive strategies, such as the reduction of indoor allergens, educational programs on asthma management, inhaler device use and other methods. 

## Figures and Tables

**Figure 1 children-09-01788-f001:**
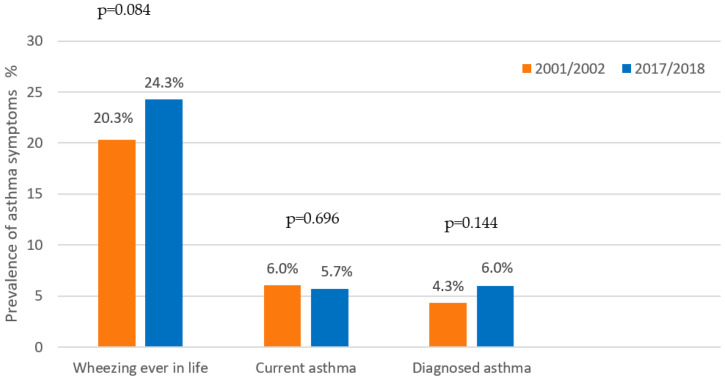
Comparison of asthma symptoms and diagnosis prevalence, 2002–2018.

**Figure 2 children-09-01788-f002:**
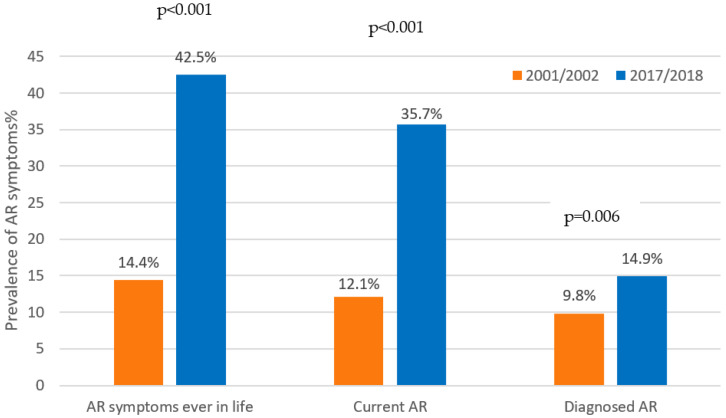
Significant rise in the self-reported prevalence of AR symptoms and diagnoses, 2002–2018.

**Figure 3 children-09-01788-f003:**
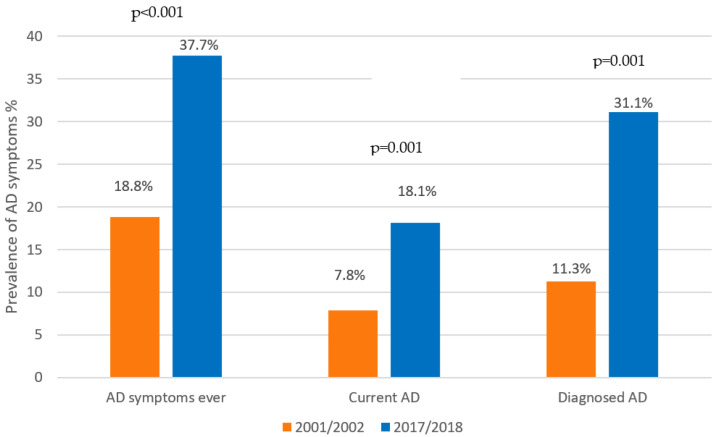
Significant rise in the self-reported prevalence of AD symptoms and diagnoses, 2002–2018.

**Figure 4 children-09-01788-f004:**
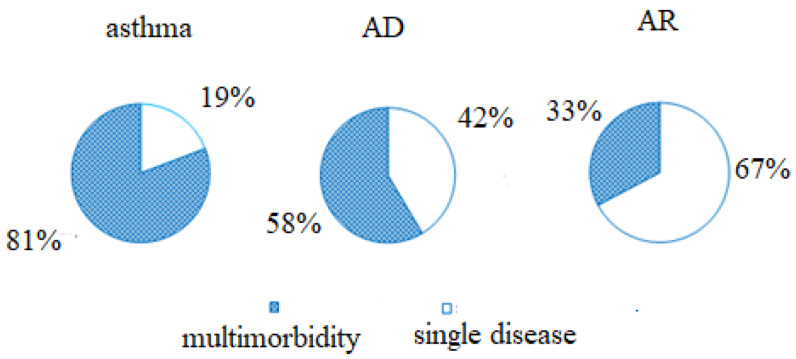
Ratio of single versus multiple disease symptoms.

**Figure 5 children-09-01788-f005:**
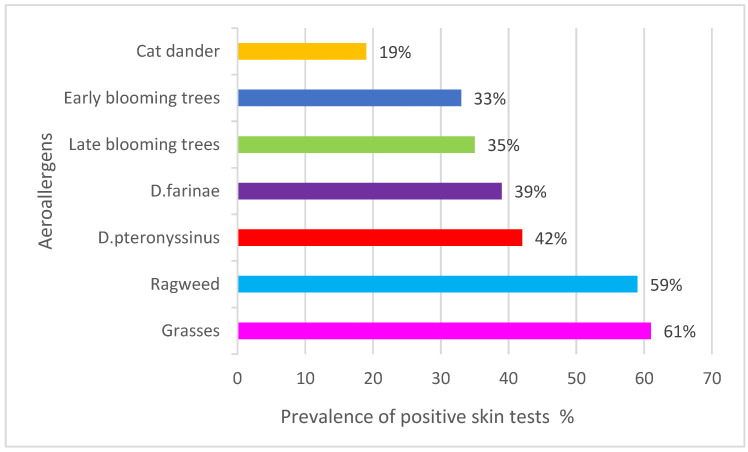
Allergic sensitisation profile obtained by SPTs in the group of 7–14-year-old children, 2018.

**Figure 6 children-09-01788-f006:**
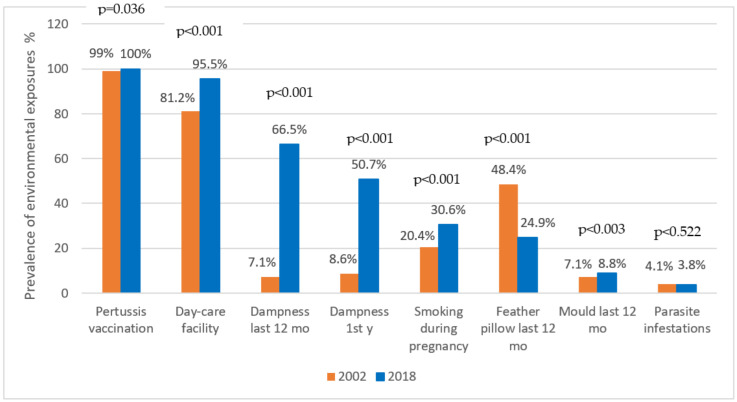
Changes in single-environmental exposures, 2002–2018.

**Table 1 children-09-01788-t001:** Epidemiological definition of asthma, allergic rhinitis (AR) and atopic dermatitis (AD).

Allergic Conditions	
Wheezing ever-in-a-lifetime	Has your child ever had wheezing in his life?
Current asthma *	Has your child had wheezing during the last 12 months in the absence of a cold?
Self-reported diagnosed asthma	Has your child ever had asthma?
AR symptoms ever-in-a-lifetime	Has your child ever had a runny nose in the absence of a cold?
Current AR *	Has your child had a runny nose without a cold during last 12 months?
Self-reported diagnosed AR	Has your child ever had allergic rhinitis?
AD symptoms ever-in-a-lifetime	Has your child ever had an itchy rash which was coming and going for at least six months?
Current AD *	Has your child had an itchy rash which was coming and going during last 12 months?
Self-reported diagnosed AD	Has your child ever had atopic dermatitis?

* Current symptoms represent the self-reported prevalence of the disease. AR—allergic rhinitis, AD—atopic dermatitis.

**Table 2 children-09-01788-t002:** Family history and environmental exposure in the study population in 2018.

Family History and Environmental Exposure	
Mother’s asthma	20 (4.4%)
Mother’s AR	87 (19.2%)
Mother’s AD	49 (10.2%)
Father’s asthma	25 (5.5%)
Father’s AR	69 (15.2%)
Father’s AD	28 (6.2%)
Attending day care	425 (95.5%)
Breastfeeding	414 (91.8%)
Pertussis vaccination	449 (100%)
Parasite infestations	17 (3.8%)
Dog in the 1st year	42 (9.3%)
Dog during the last 12 months	97 (21.4%)
Cat in the 1st year	31 (6.8%)
Cat during the last 12 months	43 (9.5%)
Tobacco smoke during pregnancy	139 (30.6%)
Home dampness in the 1st year	230 (50.7%)
Home dampness during the last 12 months	302 (66.5%)
Mould in the 1st year	36 (7.9%)
Mould during the last 12 months	40 (8.8%)
Feather pillows the last 12 months	113 (24.9%)
Hamburger consumption	
Never	343 (75.5%)
<once/week	65 (14%)
1–2x/week	40 (8.8%)
3–6x/week	3 (0.6%)
Once/day	3 (0.6%)

**Table 3 children-09-01788-t003:** Associations of family history and environmental factors with allergic symptoms.

	Wheezing Ever-in-a-Lifetime	Wheezing during the Last 12 Months	Diagnosed Asthma	AR Ever-in-a-Lifetime	AR during Last 12 Months	Diagnosed AR	AD Ever-in-a-Lifetime	AD during the Last 12 Months	Diagnosed AD
Mother’s asthma	OR 1.612 95% CI 0.448–5.803, *p* = 0.465	OR 1.555, 95% CI 0.041–32.482, *p* = 0.933	OR 8.909, 95% CI 0.698–113. 636, *p* = 0.092	OR 1.657, 95% CI 0.507–5.418 *p* = 0.404	OR 1.488, 95% CI 0.102–21.621, *p* = 0.771	OR 2.769 95% CI 0.519–14,772 *p* = 0.233	OR 0.637, 95% CI 0.178–2.283, *p* = 0.489	OR 0.415, 95% CI 0.079–2.170, *p* = 0.297	OR 1.152, 95% CI 0.347–3.824, *p* = 0.817
Mother’s AR	OR 2.813, 95% CI 1.608–4.921, *p* = 0.001	OR 4.167, 95% CI 1.284–13.527, *p* = 0.018	OR 3.082, 95% CI 1.178–8.065, *p* = 0.022	OR 1.689, 95% CI 1.018–2.803 *p* = 0.043	OR 2.354, 95% CI 0.860–6.438 *p* = 0.095	OR 2.371 95% CI 1.257–4.473 *p* = 0.008	OR 0.941, 95% CI 0.545–1.624, *p* = 0.827	OR 1.068, 95% CI 0.493–2.314 *p* = 0.868	OR 1.327, 95% CI 0.775–2.273, *p* = 0.303
Mother’s AD	OR 1.250, 95% CI 0.530–2.947, *p* = 0.610	OR 0.483, 95% CI 0.038–6.211, *p* = 0.577	OR 0.105, 95% CI 0.010–1.107, *p* = 0.061	OR 0.987, 95% CI 0.461–2.112 *p* = 0.973	OR 1.323, 95% CI 0.262–6.692 *p* = 0.735	OR 0.418 95% CI 0.118–1.481 *p* = 0.177	OR 5.124, 95% CI 2.199–11.943, *p* = 0.001	OR 1.922, 95% CI 0.749–4.93, *p* = 0.174	OR 2.752, 95% CI 1.275–5.943, *p* = 0.010
Father’s asthma	OR 2.372, 95% CI 0.890–6.327, *p* = 0.084	OR 0.516, 95% CI 0.053–5.069, *p* = 0.570	OR 2.128, 95% CI 0.506–8.950. *p* = 0.303	OR 1.559, 95% CI 0.645–3.767 *p* = 0.324	OR 1.433, 95% CI 0.250–8.223 *p* = 0.686	OR 1.744 95% CI 0.572–5.315 *p* = 0.328	OR 2.516, 95% CI 0.976–6.485 *p* = 0.056	OR 0.748, 95% CI 0.205–2.734, *p* = 0.661	OR 1.123, 95% CI 0.433–2.912, *p* = 0.812
Father’s AR	OR 2.527, 95% CI 1.338–4.773, *p* = 0.004	OR 1.696, 95% CI 0.397–7.248, *p* = 0.476	OR 5.595, 95% CI 2.068–15.132, *p* = 0.001	OR 1.497, 95% CI 0.844–2.654 *p* = 0.167	OR 1.477, 95% CI 0.512–4.257 *p* = 0.470	OR 2.841, 95% CI 1.414–5.706 *p* = 0.003	OR 1.983, 95% CI 1.087–3.617, *p* = 0.026	OR 1.572, 95% CI 0.703–3.516, *p* = 0.271	OR 2.095, 95% CI 1.159–3.787, *p* = 0.014
Father’s AD	OR 1.547, 95% CI 0.615–3.892, *p* = 0.354	OR 0.278, 95% CI 0.028–2.801, *p* = 0.277	OR 0.303, 95% CI 0.033–2.755, *p* = 0.289	OR 1.140, 95% CI 0.493–2.633 *p* = 0.759	OR 1.275, 95% CI 0.282–5.770 *p* = 0.752	OR 1.188 95% CI 0.429–3.289 *p* = 0.740	OR 2.083, 95% CI 0.849–5.108, *p* = 0.109	OR 2.894, 95% CI 0.848–9.878, *p* = 0.090	OR 2.362, 95% CI 0.993–5.622, *p* = 0.052
Attending day care	OR 0.317, 95% CI 0.686–16.038, *p* = 0.136	OR 1.098, 95% CI 0.043–28.357, *p* = 0.955	OR 0.420, 95% CI 0.070–2.525, *p* = 0.343	OR 1.745, 95% CI 0.608–5.004 *p* = 0.300	OR 2.182, 95% CI 0.303–15.710, *p* = 0.439	OR 4.173 95% CI 0.495–35.129 *p* = 0.300	OR 3.033, 95% CI 0.898–10.245, *p* = 0.074	OR 0.371, 95% CI 0.047–2.951, *p* = 0.349	OR 1.724, 95% CI 0.515–5.768, *p* = 0.377
Dog in the 1st year	OR 0.834, 95% CI 0.351–1.980, *p* = 0.680	OR 7.548, 95% CI 0.989–57.647, *p* = 0.051	OR 3.455, 95% 0.988–12.079, *p* = 0.052	OR 1.003, 95% CI 0.479–2.100 *p* = 0.994	OR 0.853 95% CI 0.172–4.236 *p* = 0.846	OR 0.550 95% CI 0.171–1.776 *p* = 0.318	OR 0.663, 95% CI 0.289–1.521 *p* = 0.332	OR 2.312, 95% CI 0.524–10.199, *p* = 0.269	OR 1.001, 95% CI 0.436–2.299, *p* = 0.998
Dog during the last 12 months	OR 2.033, 95% CI 1.127–3.667, *p* = 0.018	OR 0.812, 95% CI 0.208–3.163, *p* = 0.764	OR 1.961, 95% CI 0.682–5.639, *p* = 0.211	OR 1.200, 95% CI 0.712–2.021 *p* = 0.493	OR 1.677 95% CI 0.556–5.052 *p* = 0.359	OR 1.441 95% CI 0.707–2.937 *p* = 0.315	OR 0.837, 95% CI 0.474–1.478 *p* = 0.539	OR 1.545, 95% CI 0.604–3.95, *p* = 0.364	OR 0.764, 95% CI 0.425–1.373 *p* = 0.368
Cat in the 1st year	OR 0.525, 95% CI 0.165–1.673, *p* = 0.276	OR 12.273, 95% CI 0.823–183.006, *p* = 0.069	OR 0.883, 95% CI 0.148–5.259, *p* = 0.892	OR 1.093, 95% CI 0.452–2.640 *p* = 0.844	OR 1.880 95% CI 0.406–8.717 *p* = 0.420	OR 1.635 95% CI 0.481–5.561 *p* = 0.431	OR 0.831, 95% CI 0.312–2.215 *p* = 0.712	OR 0.520, 95% CI 0.081–3.345, *p* = 0.491	OR 0.792, 95% CI 0.286–2.188, *p* = 0.652
Cat during the last 12 months	OR 1.446, 95% CI 0.622–3.362, *p* = 0.392	OR 1.672, 95% CI 0.256–10.931 *p* = 0.592	OR 1.593, 95% CI 0.336–7.540, *p* = 0.557	OR 0.737, 95% CI 0.350–1.550 *p* = 0.421	OR 0.277 95% CI 0.087–0.883 *p* = 0.030	OR 0.892 95% CI 0.307–2.59 *p* = 0.833	OR 0.827, 95% CI 0.372–1.836 *p* = 0.640	OR 1.500, 95% CI 0.412–5.462, *p* = 0.539	OR 1.133, 95% CI 0.515–2.495, *p* = 0.756
Tobacco smoke during pregnancy	OR 1.253, 95% CI 0.743–2.112, *p* = 0.398	OR 0.805, 95% CI 0.207–3.137, *p* = 0.755	OR 0.718, 95% CI 0.244–2.114, *p* = 0.548	OR 1.235, 95% CI 0.799–1.910 *p* = 0.343	OR 1.584, 95% CI 0.659–3.806 *p* = 0.304	OR 1.511 95% CI 0.816–2.797 *p* = 0.189	OR 0.934, 95% CI 0.586–1.486, *p* = 0.772	OR 1.228, 95% CI 0.590–2.557, *p* = 0.583	OR 1.073, 95% CI 0.665–1.729, *p* = 0.773
Home dampness in the 1st year	OR 0.672, 95% CI 0.368–1.226, *p* = 0.195	OR 3.734, 95% CI 0.877–15.903 *p* = 0.075	OR 1.351, 95% CI 0.417–4.379, *p* = 0.616	OR 1.034, 95% CI 0.636–1.678 *p* = 0.894	OR 1.015, 95% CI 0.409–2.524 *p* = 0.974	OR 1.071 95% CI 0.532–2.158 *p* = 0.847	OR 1.314, 95% CI 0.780–2.213, *p* = 0.304	OR0.866, 95% CI 0.401–1.968, *p* = 0.713	OR 0.942, 95% CI 0.554–1.602, *p* = 0.827
Home dampness during the last 12 months	OR 0.723, 95% CI 0.394–1.329, *p* = 0.297	OR 2.185, 95% CI 0.496–9.628, *p* = 0.302	OR 0.649, 95% CI 0.201–2.101, *p* = 0.471	OR 1.106, 95% CI 0.660–1.854 *p* = 0.703	OR 2.220 95% CI 0.866–5.688 *p* = 0.097	OR 1.063 95% CI 0.503–2.245 *p* = 0.873	OR 0.748, 95% CI 0.429–1.306, *p* = 0.307	OR 1.372, 95% CI 0.595–3.163, *p* = 0.458	OR 0.981, 95% CI 0.556–1.733, *p* = 0.949
Mold in the 1st year	OR 0.889, 95% CI 0.333–2.376, *p* = 0.814	OR 14.505, 95% CI 1.076–195.524, *p* = 0.044	OR 1.162, 95% CI 0.166–8−148, *p* = 0.880	OR 1.098, 95% CI 0.455–2.651 *p* = 0.836	OR 3.818 95% CI 0.470–31.025 *p* = 0.210	OR 1.909 95% CI 0.542–6.722 *p* = 0.314	OR 1.242, 95% CI 0.486–3.173, *p* = 0.651	OR 0.315, 95% CI 0.072–1.374, *p* = 0.124	OR 0.677, 95 %CI 0.245–1.869, *p* = 0.451
Mold during the last 12 months	OR 1.242, 95% CI 0.448–3.444, *p* = 0.667	OR 2.481, 95% CI 0.280–21.998, *p* = 0.414	OR 1.194, 95% CI 0.189–7.523, *p*= 0.851	OR 0.908, 95% CI 0.400–2.063 *p* = 0.818	OR 0.507 95% CI 0.109–2.349 *p* = 0.385	OR 0.402 95% CI 0.106–1.529 *p* = 0.181	OR 1.215, 95% CI 0.519–2.846, *p* = 0.654	OR 3.067, 95% CI 0.818–11.491, *p* = 0.096	OR 1.609, 95% CI 0.672–3.851, *p* = 0.286

## Data Availability

Data are available from the authors.

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
