# Peer review of "Divergent Trends in the Prevalence of Children’s Asthma, Rhinitis and Atopic Dermatitis and Environmental Influences in the Urban Setting of Zagreb, Croatia"

_children, 2022, doi:10.3390/children9121788_

Round 1

Reviewer 1 Report

I believe this manuscript could be interesting to the readers of Children and it is worth to be published. However, there are some comments and suggestions to authors:

·         Materials and Methods

ü  The authors reported the very low response rate in the repeated study. It is important to state how much it really was and to compare it with the response rate obtained in the first study.

ü  Neither is clearly stated nor cited which set of questions and/or questionnaires about environmental exposure and family history were used. ISAAC questionnaires?

ü  Assessment of allergic multimorbidities should be mentioned. In general, all the variables which were analyzed and reported as results should be specified in the section "Materials and Methods" at first.

·         Discussion

ü  I believe that the problem of the low response rate (not reported exactly) and the modest sample size of 454 subjects in the second study (far less than 1000 subjects as the methodological minimum recommended by ISAAC) should be mentioned and discussed as a possible methodological weakness of the second study that may partly affect the results and the conclusions of this research. For example, a reduced response rate can affect the representativeness of the sample resulting in a possibility that parents of children suffering from allergic diseases will be more motivated to participate in the study than the parents of healthy children. This can especially apply to parents of children who suffer from atopic dermatitis and allergic rhinitis - allergic diseases in which quality of life could be more affected than in patients with asthma.

ü  Methodological discrepancy between the first and the second study related to the age of subjects (10-11y vs. 7-14y) who were skin prick tested could also affect final conclusions. For example, it is expected that with the increasing age of tested children prevalence of sensitization to pollen allergens increases too. I believe these moments should be also the subject of discussion.

I believe this manuscript could be interesting to the readers of Children and it is worth to be published. However, there are some comments and suggestions to authors:

·         Materials and Methods

ü  The authors reported the very low response rate in the repeated study. It is important to state how much it really was and to compare it with the response rate obtained in the first study.

ü  Neither is clearly stated nor cited which set of questions and/or questionnaires about environmental exposure and family history were used. ISAAC questionnaires?

ü  Assessment of allergic multimorbidities should be mentioned. In general, all the variables which were analyzed and reported as results should be specified in the section "Materials and Methods" at first.

·         Discussion

ü  I believe that the problem of the low response rate (not reported exactly) and the modest sample size of 454 subjects in the second study (far less than 1000 subjects as the methodological minimum recommended by ISAAC) should be mentioned and discussed as a possible methodological weakness of the second study that may partly affect the results and the conclusions of this research. For example, a reduced response rate can affect the representativeness of the sample resulting in a possibility that parents of children suffering from allergic diseases will be more motivated to participate in the study than the parents of healthy children. This can especially apply to parents of children who suffer from atopic dermatitis and allergic rhinitis - allergic diseases in which quality of life could be more affected than in patients with asthma.

ü  Methodological discrepancy between the first and the second study related to the age of subjects (10-11y vs. 7-14y) who were skin prick tested could also affect final conclusions. For example, it is expected that with the increasing age of tested children prevalence of sensitization to pollen allergens increases too. I believe these moments should be also the subject of discussion.

I believe this manuscript could be interesting to the readers of Children and it is worth to be published. However, there are some comments and suggestions to authors:

·         Materials and Methods

ü  The authors reported the very low response rate in the repeated study. It is important to state how much it really was and to compare it with the response rate obtained in the first study.

ü  Neither is clearly stated nor cited which set of questions and/or questionnaires about environmental exposure and family history were used. ISAAC questionnaires?

ü  Assessment of allergic multimorbidities should be mentioned. In general, all the variables which were analyzed and reported as results should be specified in the section "Materials and Methods" at first.

·         Discussion

ü  I believe that the problem of the low response rate (not reported exactly) and the modest sample size of 454 subjects in the second study (far less than 1000 subjects as the methodological minimum recommended by ISAAC) should be mentioned and discussed as a possible methodological weakness of the second study that may partly affect the results and the conclusions of this research. For example, a reduced response rate can affect the representativeness of the sample resulting in a possibility that parents of children suffering from allergic diseases will be more motivated to participate in the study than the parents of healthy children. This can especially apply to parents of children who suffer from atopic dermatitis and allergic rhinitis - allergic diseases in which quality of life could be more affected than in patients with asthma.

ü  Methodological discrepancy between the first and the second study related to the age of subjects (10-11y vs. 7-14y) who were skin prick tested could also affect final conclusions. For example, it is expected that with the increasing age of tested children prevalence of sensitization to pollen allergens increases too. I believe these moments should be also the subject of discussion.

I believe this manuscript could be interesting to the readers of Children and it is worth to be published. However, there are some comments and suggestions to authors:

·         Materials and Methods

ü  The authors reported the very low response rate in the repeated study. It is important to state how much it really was and to compare it with the response rate obtained in the first study.

ü  Neither is clearly stated nor cited which set of questions and/or questionnaires about environmental exposure and family history were used. ISAAC questionnaires?

ü  Assessment of allergic multimorbidities should be mentioned. In general, all the variables which were analyzed and reported as results should be specified in the section "Materials and Methods" at first.

·         Discussion

ü  I believe that the problem of the low response rate (not reported exactly) and the modest sample size of 454 subjects in the second study (far less than 1000 subjects as the methodological minimum recommended by ISAAC) should be mentioned and discussed as a possible methodological weakness of the second study that may partly affect the results and the conclusions of this research. For example, a reduced response rate can affect the representativeness of the sample resulting in a possibility that parents of children suffering from allergic diseases will be more motivated to participate in the study than the parents of healthy children. This can especially apply to parents of children who suffer from atopic dermatitis and allergic rhinitis - allergic diseases in which quality of life could be more affected than in patients with asthma.

ü  Methodological discrepancy between the first and the second study related to the age of subjects (10-11y vs. 7-14y) who were skin prick tested could also affect final conclusions. For example, it is expected that with the increasing age of tested children prevalence of sensitization to pollen allergens increases too. I believe these moments should be also the subject of discussion.

Author Response

Please see the attachment bellow. 

Reviewer 2 Report

Allergic diseases are an important field of research, owing to ever-increasing social costs as well as their impact on the patients' quality of life. The data offered by the Authors in this paper is useful in clarifying the epidemiologic and physiopathological characteristics of asthma, atopic dermatitis and allergic rhinitis in young Croatian patients.

While there is undoubledly merit in the paper, I have some questions:

Ln 69: can the Authors elucidate on the reasons for such low enrolment in the study versus the previous initiative?
Ln 80: how were environmental factors determined? Some choices appear rather peculiar (e.g. hamburger consumption). The paper does not include a full list - perhaps a copy of the study questionnaire should be included as a supplement?
Ln 92: was the follow-up study cohort sex- and age-matched with the 2001 one? 
Ln 129-130: "other age groups", perhaps this new cohort should be better defined? How many males-females? How were the young patients selected? Furthermore, the full skin prick test (SPT) panel is not specified, leaving out some crucial allergens (e.g. dog fur)
Ln 141: what age range attends daycare? What was the numerosity in the 2001 versus 2018 cohorts? Furthermore, please note that the higher exposure to mould could be related to differences in the selected cohorts (for example socioeconomic differences)
Ln 215: dog allergen was investigated via questionnaire only, not via SPTs, am I mistaken?

Furthermore, spelling mistakes are present throughout the document (e.g. table 2's header).

Author Response

Please see the attachment bellow
